# Intuition and Symmetries in Electromagnetism: Eigenstates of Four Antennas

Hamdi Bilel *[ID] and Aguili Taoufik

Communication System Laboratory Sys'Com, National Engineering School of Tunis (ENIT), University of Tunis El Manar, BP 37, Le Belvédère, Tunis 1002, Tunisia
* Correspondence: hbilel.enit@gmail.com; Tel.: +33-605903234

**Abstract:** Symmetries play an essential role in the field of physics. In this paper, we examine the relationship between the eigen-amplitudes of four (2 × 2) symmetrical antennas and the symmetry of the amplitudes of their sources (excitations) using mirroring effects. In our case, we find that changing mirrors using symmetry is identical to the point group theory. By exploiting the symmetry problem, we can show the advantage of reducing the size of the analysis domain, at least by a factor of two or more (2, 4, and 8. . . etc.) (depending on the problem). Several simulation examples have been developed by the MoM-GEC and HFSS to validate this approach.

**Keywords:** symmetries; antennas; perfect electric conductor (PEC) walls; perfect magnetic conductor (PMC) walls; mirroring effects; eigen states; superposition; phase-shift; S-parameters; HFSS; MoM-GEC





## 1. Introduction

Various commercial software packages are interested in studying symmetry problems, such as HFSS [1], CST [2], Cadence [3], Keysight [4], Comsol [5], Optiwave [6] . . . etc. In addition, symmetry analysis can be employed successfully in many aspects of electromagnetic theory. It can be implemented in the differential, integral, variational and matrix description of electromagnetic phenomenons. Therefore, Maxwell's equations have a very high symmetry. So, all exact solutions of the wave equations in Cartesian, cylindrical and spherical coordinates are derived from the symmetry of these differential equations [7]. In addition, several bibliographies have focused on the applications of symmetry in physics and in particular in electromagnetism. The properties of an antenna array are related to the type of symmetry that prevails in it. Group point theory is a systematic [8,9], but not always convenient, tool to exploit this symmetry, in particular when it comes to finding the eigenvectors and eigenvalues of an operator [10–12] .

In general, symmetry properties are used to simplify the solution of electromagnetic phased array problems by reducing the domain analysis, where each system obeying these (symmetry) characteristics undergoes several transformations that facilitate their rewriting [12–14]. In this context, other special cases are the result of the symmetry intuition (these are rare cases in electromagnetism, as explained in [15]). This is why, in this paper, we propose a special case of eigenstates of four antennas (extendable to a subarrays symmetry problem) given as the result of a superposition of symmetry states that are established by combinations of axial symmetries of a perfect electric conductor (PEC) wall and a perfect magnetic conductor (PMC) wall [13,14,16]. A technique of phase shift between two sources based on the calculation of S-parameters is used to determine the different combinations of symmetries [17]. Note that a superposition of symmetry states equivalent to the sources is possible with the S parameters. This approach can be applied to several numerical methods in electromagnetics [13,14], including the method of moments simplified by equivalent circuits (MoM GEC) [18].

This article is divided into four parts. First, we start by formulating the problem by explaining how to construct the symmetries and how to use them to establish the superposition theorem to produce an eigenstate of four antennas (compared to the symmetry group point theory). Next, a phase shift technique between these sources is used to reveal these symmetries. Then, results are presented based on the MoM-GEC and HFSS to clarify these symmetries. An advantage of reducing the analysis domain of the problem is discussed. Finally, a perspective is provided and conclusions are drawn.

## 2. Problem Formulation

### 2.1. Setting Up of the Problem

Let $S_1$, $S_2$, $S_3$ and $S_4$ be four sources of self-amplitudes $E_1$, $E_2$, $E_3$ and $E_4$ of four antennas as given in Figure 1 [13,14] . Each eigenstate of symmetry (under the condition of the mirror effect) has an amplitude $\tilde{E}_1$, $\tilde{E}_2$, $\tilde{E}_3$ and $\tilde{E}_4$, as we are going to explain [13,14,19]. **\*Proper states:**

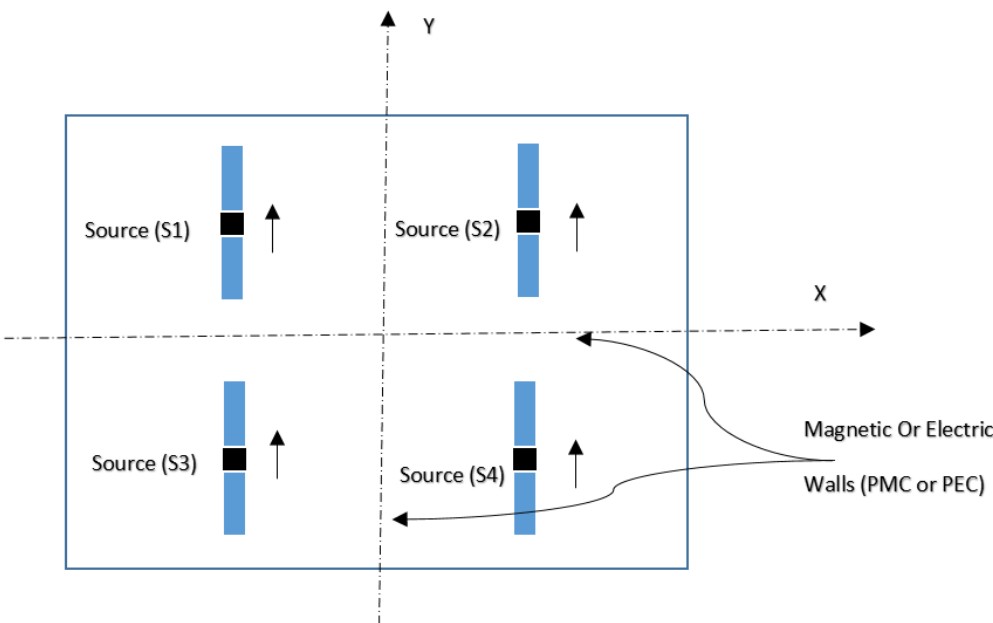

**Figure 1.** Four antenna configuration with a combination of electrical and magnetic symmetries [13,14].

Depending on the direction (ox): we can put up a magnetic wall or an electric wall. Similarly for the direction (oy), we can set up a magnetic wall or an electric wall, as described in Figure 1 [16,19–34] .

The combination between the two axes (ox) and (oy) allows us to establish four states of amplitude mirroring, which can be summarized as follows (see Table 1 of [13,14]) [12–15,20–22,35]:

| States | Walls | Amplitudes of sources |
|---|---|---|
| 1 | electric (ox)\magnetic (oy) | 1 1 1 1 |
| 2 | magnetic (ox)\magnetic (oy) | 1 1 −1 −1 |
| 3 | electric (ox)\electric (oy) | 1 −1 1 −1 |
| 4 | magnetic (ox)\electric (oy) | 1 −1 −1 1 |

Similar to the theory of point group symmetry [8], the amplitudes of the sources are defined as the tables (Tables 9.3. and 9.4. of [8]), extensible to other types of symmetries (given in the same reference [8]). Note that the type of these symmetries is explained in the same tables. For example in our case, the orthogonal excitations are of group $C_{2v}$ (see Figure 9.4. of [8]).

The character table for the $C_{2v}$ symmetry point group is given below (identical to the arrangement of the source amplitudes):

| States | $C_{2v}$ | $E$ | $C_2^z$ | $\sigma_v(x,z)$ | $\sigma_v'(y,z)$ |
|---|---|---|---|---|---|
| 1 | $u_1$ | 1 | 1 | 1 | 1 |
| 2 | $u_2$ | 1 | 1 | $-1$ | $-1$ |
| 3 | $u_3$ | 1 | $-1$ | 1 | $-1$ |
| 4 | $u_4$ | 1 | $-1$ | $-1$ | 1 |
| | $h = 4$ | $l_1 = 1$ | $l_2 = 1$ | $l_3 = 1$ | $l_4 = 1$ |

The covering operations of the group $C_{2v}$ are: the identity $E$, $C_2$ rotation around the z-axis, $\sigma_v$ plane of symmetry about the x-z-plane and $\sigma_v'$ plane of symmetry about the y-z-plane.

The symmetry based on the combinations of electric-magnetic walls [13,14] is identical to the point group symmetry theory [8]. Both verify the case of four antennas symmetry [8,13,14].

Let us now return to the amplitudes of the sources in the first table, By normalizing the states, we have:

$$u_1 = \begin{pmatrix} \frac{1}{2} \\ \frac{1}{2} \\ \frac{1}{2} \\ \frac{1}{2} \end{pmatrix}, u_2 = \begin{pmatrix} \frac{1}{2} \\ \frac{1}{2} \\ -\frac{1}{2} \\ -\frac{1}{2} \end{pmatrix}, u_3 = \begin{pmatrix} \frac{1}{2} \\ -\frac{1}{2} \\ \frac{1}{2} \\ -\frac{1}{2} \end{pmatrix} \text{ and } u_4 = \begin{pmatrix} \frac{1}{2} \\ -\frac{1}{2} \\ -\frac{1}{2} \\ \frac{1}{2} \end{pmatrix}$$

$u_1$, $u_2$, $u_3$ and $u_4$ are orthonormal vectors.

Then, using the theorem of superposition, any state can be written [13,14]:

$$E = \sum_{i=1} E_i V_i = \sum_{i=1} \tilde{E}_i U_i \tag{1}$$

where

$$V_1 = \begin{pmatrix} 1 \\ 0 \\ 0 \\ 0 \end{pmatrix}, V_2 = \begin{pmatrix} 0 \\ 1 \\ 0 \\ 0 \end{pmatrix}, V_3 = \begin{pmatrix} 0 \\ 0 \\ 1 \\ 0 \end{pmatrix} \text{ and } V_4 = \begin{pmatrix} 0 \\ 0 \\ 0 \\ 1 \end{pmatrix}$$

$$\Rightarrow E = \underbrace{\begin{matrix} V_1 & V_2 & V_3 & V_4 \\ \begin{pmatrix} 1 & 0 & 0 & 0 \\ 0 & 1 & 0 & 0 \\ 0 & 0 & 1 & 0 \\ 0 & 0 & 0 & 1 \end{pmatrix} \end{matrix}}_{\text{Proper vectors}} \begin{pmatrix} E_1 \\ E_2 \\ E_3 \\ E_4 \end{pmatrix}$$

$$= \underbrace{\begin{pmatrix} E_1 \\ E_2 \\ E_3 \\ E_4 \end{pmatrix}}_{\text{Eigen amplitudes of the antennas}} \text{, so,}$$

$$E = \begin{pmatrix} E_1 \\ E_2 \\ E_3 \\ E_4 \end{pmatrix} = \frac{1}{2} \begin{matrix} u_1 & u_2 & u_3 & u_4 \\ \begin{pmatrix} 1 & 1 & 1 & 1 \\ 1 & 1 & -1 & -1 \\ 1 & -1 & 1 & -1 \\ 1 & -1 & -1 & 1 \end{pmatrix} \end{matrix} \begin{pmatrix} \tilde{E}_1 \\ \tilde{E}_2 \\ \tilde{E}_3 \\ \tilde{E}_4 \end{pmatrix} = P\tilde{E}$$

The passage matrix P is unitary $\Rightarrow (P^{-1} = P^t)$.

$\Rightarrow \tilde{E} = P^{-1}E = P^t E$

$\Rightarrow$

$$\underbrace{E}_{\text{Amplitudes of the antennas}} = P \underbrace{\tilde{E}}_{\text{Amplitudes of the states (sources in symmetries)}}$$

Each antenna self-amplitude in the total configuration of four antennas is written as the superposition of the symmetry amplitudes (states) (all the combinations of symmetry between electric and magnetic walls are considered) [13,14].

### 2.2. Symmetries and Phases

In this case, we consider only the case of two sources (a source and its image) having an even or odd symmetry relation (in the presence of magnetic or electric planes) [13,16,26–34]. According to the theorem of image explained in [16,20–24,31–34], we can simply count the phases established between two symmetrical or anti-symmetrical sources (in phase or in phase opposition), as indicated in Figure 2 [13].

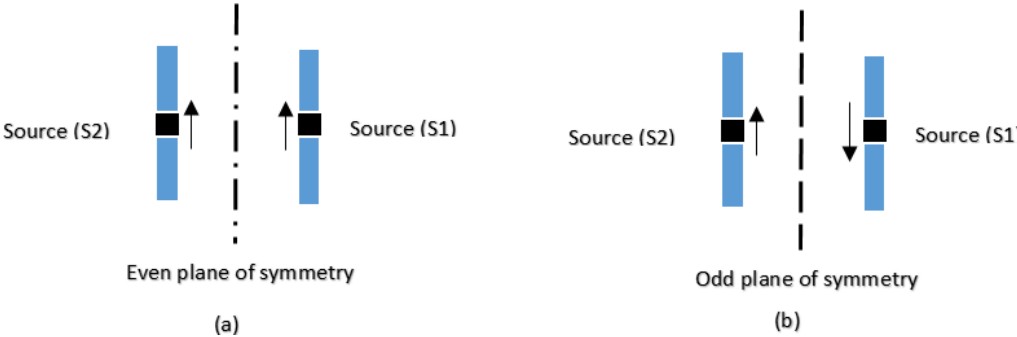

**Figure 2.** Symmetry and antisymmetry between two sources [13].

In the case of even symmetry, the phase shift established between two sources was:

$$\Phi_{S12} = 0[2\pi] = 2k\pi \Rightarrow 2 \text{ sources are in phase} \tag{2}$$

Inversely, in the case of odd symmetry, the phase shift was :

$$\Phi_{S12} = (2k+1)[\pi] = (2k+1)\pi \tag{3}$$
$$\Rightarrow 2 \text{ sources are in phase opposition}$$

According to Appendix A [17], $S$ can be written as:

$$[S] = \begin{pmatrix} S_{11}e^{-j\Phi1} & S_{12}e^{-j(\Phi2+\Phi1)} \\ S_{21}e^{-j(\Phi2+\Phi1)} & S_{22}e^{-j\Phi2} \end{pmatrix} \tag{4}$$

∗ **Special cases:**
◇ Case of even symmetry: if we fix $\Phi1 = 0 \Rightarrow \Phi2 = 0$.
So,

$$[S] = \begin{pmatrix} S_{11} & S_{12} \\ S_{21} & S_{22} \end{pmatrix} \tag{5}$$

◇ Case of odd symmetry: if we fix $\Phi1 = 0 \Rightarrow \Phi2 = \pi$ and vice versa if $\Phi1 = \pi \Rightarrow \Phi2 = 0$.
⇒

$$[S] = \begin{pmatrix} S_{11} & S_{12}e^{-j\pi} \\ S_{21}e^{-j\pi} & S_{22}e^{-j\pi} \end{pmatrix} \tag{6}$$

◇ In the case of an arbitrary phase shift and an odd symmetry: we define any phase $\Phi1$ will automatically add a phase shift of $\pi$ to $\Phi2$ :

$\Rightarrow \Phi2 = \Phi1 + \pi \Rightarrow \Phi2 - \Phi1 = \pi$ (Note that this case is verified under HFSS with the phase commands (e.g., $Arg(S_{21})$). Finally, the matrix S is written :

$$[S] = \begin{pmatrix} S_{11}e^{-j\Phi1} & S_{12}e^{-j(2\Phi1+\pi)} \\ S_{21}e^{-j(2\Phi1+\pi)} & S_{22}e^{-j(\Phi1+\pi)} \end{pmatrix} \tag{7}$$

We can generalize these different cases to study the configurations of the previous section (two by two between the four antennas). It is now possible to simulate this problem using commercial software (such as HFSS and CST) or other software.

Finally, we can generalize the case of [S] parameters adapted to four-antennas structure in each symmetry state (in the presence of all combinations of magnetic and electrical walls), which is written as follows [13,14,17]:

$$[S] = \begin{pmatrix} S_{11}e^{-j\Phi1} & S_{12}e^{-j(\Phi2-\Phi1)} & S_{13}e^{-j(\Phi3-\Phi1)} & S_{14}e^{-j(\Phi4-\Phi1)} \\ S_{21}e^{-j(\Phi2-\Phi1)} & S_{22}e^{-j\Phi2} & S_{23}e^{-j(\Phi3-\Phi2)} & S_{24}e^{-j(\Phi4-\Phi2)} \\ S_{31}e^{-j(\Phi3-\Phi1)} & S_{32}e^{-j(\Phi3-\Phi2)} & S_{33}e^{-j(\Phi3)} & S_{34}e^{-j(\Phi4-\Phi3)} \\ S_{41}e^{-j(\Phi4-\Phi1)} & S_{42}e^{-j(\Phi4-\Phi2)} & S_{43}e^{-j(\Phi4-\Phi3)} & S_{44}e^{-j(\Phi4)} \end{pmatrix} \tag{8}$$

The symbols $\Phi1$, $\Phi2$, $\Phi3$ and $\Phi4$ are parameters that depend on the nature of the symmetry (odd\even) used at each configuration. The particular cases of odd-even symmetries of Equation (8) will be treated in the same way as the case of two sources (see Equations (5)–(7)). Algebraically, the superposition of the S parameters is possible as the sources amplitudes, as proposed in the previous section.

## 3. Results

To distinguish the different cases of symmetry, it is necessary to use the phases of the physical quantities J, E, H and the coupling parameters S, Z, Y, . . . etc. In our case, we used the MoM-GEC and HFSS as simulation tools (see Figure 3). In this context, several results were shown to validate this approach. First, we validated the MoM GEC and HFSS on the input impedance of the planar dipole antenna used in the four antennas configuration, as indicated in Figure 4 [18]. After that, a validation based on the boundary conditions of four antennas was proven by the surface currents described by the guide modes and test functions, solved with MoM GEC, as given in Figures 5 and 6 [18].

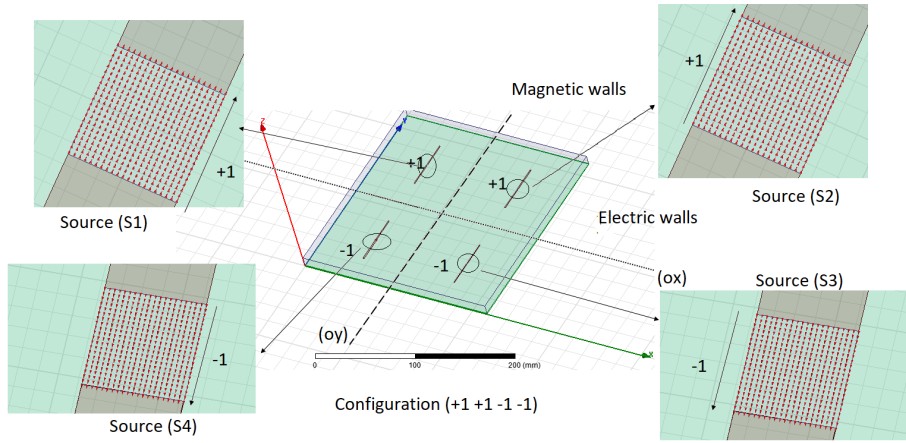

**Figure 3.** Example of a four symmetric antennas configuration with amplitudes of $11-1-1$ (under HFSS): (ox) electrical walls, (oy) magnetic walls.

A limitation with the MoM-GEC is shown to differentiate the different cases of symmetries. According to the MoM GEC formulation, the coupling parameters Z, Y and S are independent of the phase shifts of the excitation sources (From the coupling expression $Z = \frac{1}{A^t[Z_{i,j}]^{-1}A}$) [18]. These phase shifts will be caused only by the amplitudes of the surface

currents $J_S$ and the surface electric field $E_S$ ($J_S$ and $E_S$ are complex numbers). We were able to differentiate these different symmetries with the MoM-GEC using the surface current phase of the four-antennas structure, as given in Figures 7 and 8. It is an advantage for HFSS to show and distinguish these symmetries with the coupling parameters Z, Y and S by using the commands Arg(S), Arg(Y) and Arg(Z) (or the HFSS Phase commands).

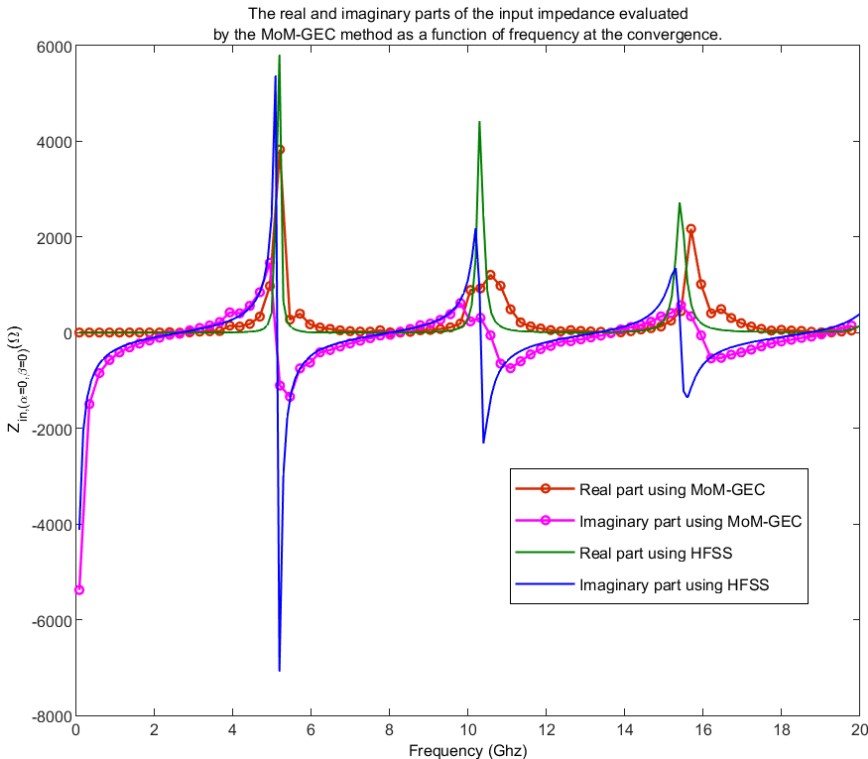

**Figure 4.** Input impedance seen at the source of one of four antennas given by the MoM GEC method and the HFSS tool (Validation) [18].

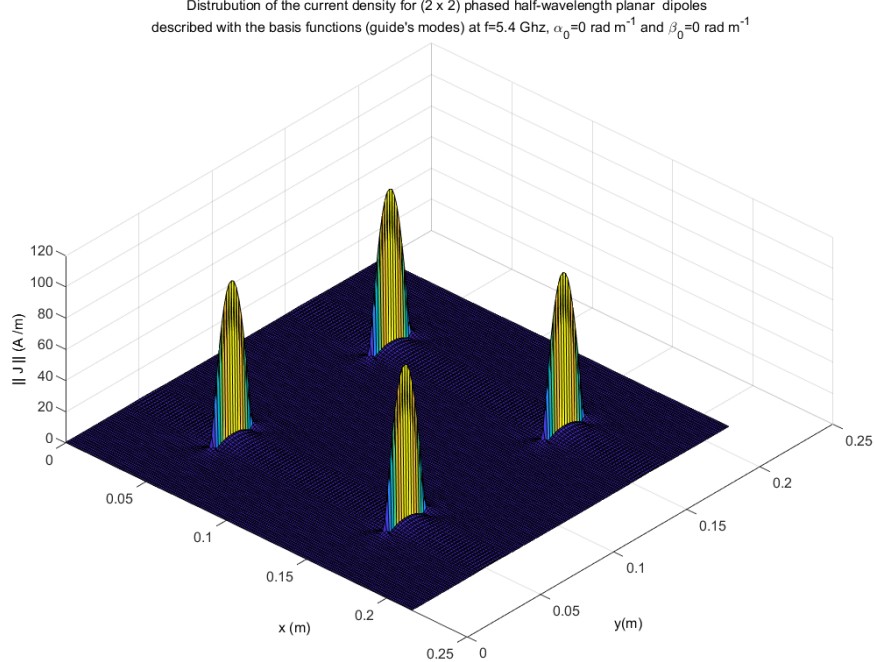

**Figure 5.** Distribution of the current density for ($2 \times 2$) phased half-wavelength planar dipoles described with the guide's modes functions at f = 5.4 Ghz, $\alpha_0 = 0$ rad m$^{-1}$ and $\beta_0 = 0$ rad m$^{-1}$ (MoM-GEC method) [18].

According to Figure 9, we considered Arg($S_{12}$) between the interaction of two sources of the configuration 1111 (two symmetrical sources in-phase) and Arg($S_{12}$) between the interaction of two sources of the configuration 11-1-1 (two anti-symmetrical sources in phase opposition). We found a phase shift of angle $\pi$ is established between Arg($S_{12}^{\text{symmetric sources}}$) and Arg($S_{12}^{\text{anti symmetric sources}}$) , at any point of the frequency band [0 20] GHz, as depicted in Figure 9.

This verifies, Arg($S_{12}^{\text{symmetric sources}}$) $-$ Arg($S_{12}^{\text{anti symmetric sources}}$) $= \pi$ (or = 180°) (also, Arg($Z_{12}^{\text{symmetric sources}}$) $-$ Arg($Z_{12}^{\text{anti symmetric sources}}$) $= \pi$) which seems to follow the equation: $\Phi_1 - \Phi_2 = \pi$ and the reasoning which follows Equation (6). In our case, we used two frequencies, 3.9948 Ghz and 11.8029 Ghz, on the whole frequency band. At each frequency, if we add or subtract their values (between Arg (S21) of two symmetrical sources and Arg (S21) of two antisymmetrical sources), we find a phase shift of 180 degrees. Note that we checked this difference on the whole frequency band. Finally, we can distinguish the cases of symmetries by the introduction of phase shift in the matrix of [S] parameters, as explained in the previous section.

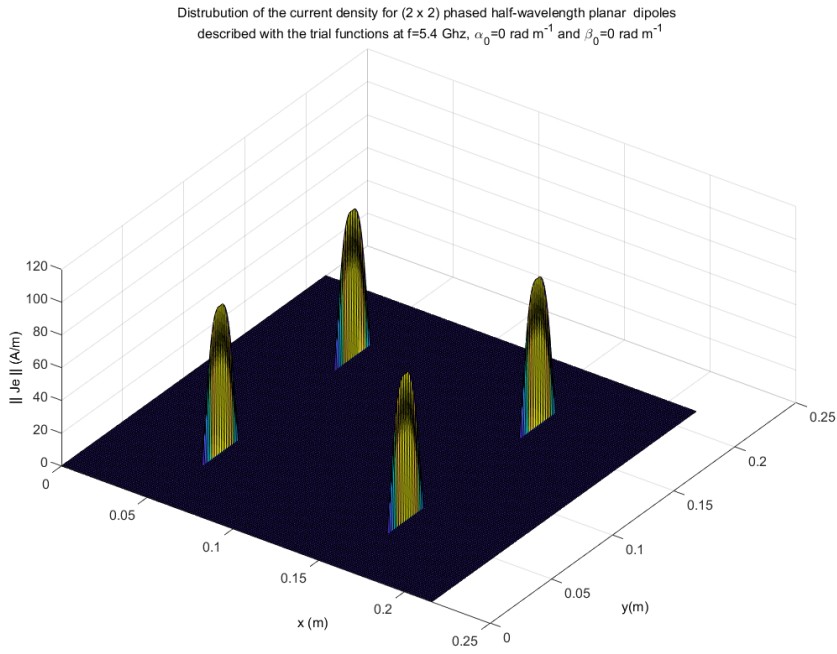

**Figure 6.** Distribution of the current density for (2 × 2) phased half-wavelength planar dipoles described with the trial functions (test functions) at f = 5.4 Ghz, $\alpha_0 = 0$ rad m$^{-1}$ and $\beta_0 = 0$ rad m$^{-1}$ (MoM-GEC method) [18].

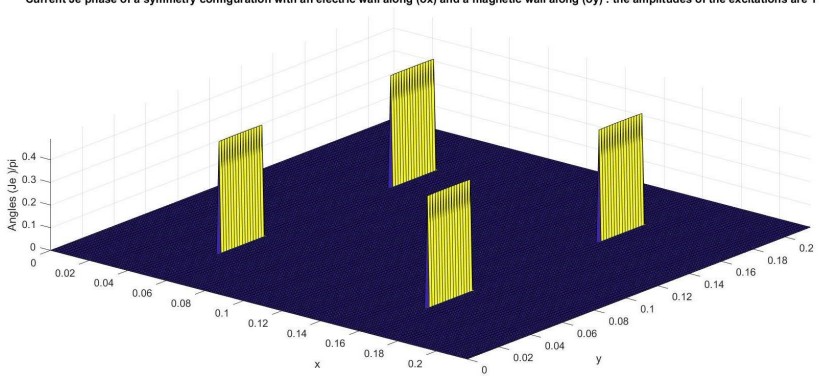

**Figure 7.** Current $J_e$'s phase (or Angle) of a symmetry configuration with an electric wall along (ox) and a magnetic wall along (oy): the amplitudes of the excitations are 1111 (MoM-GEC method).

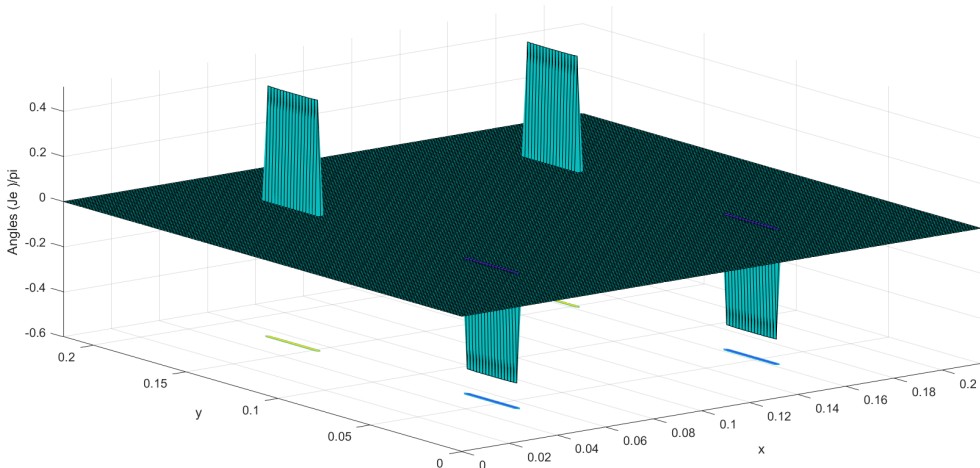

**Figure 8.** Current $J_e$'s phase (or Angle) of a symmetry configuration with a magnetic wall along (ox) and a magnetic wall along (oy): the amplitudes of the excitations are 11−1−1 (MoM-GEC method).

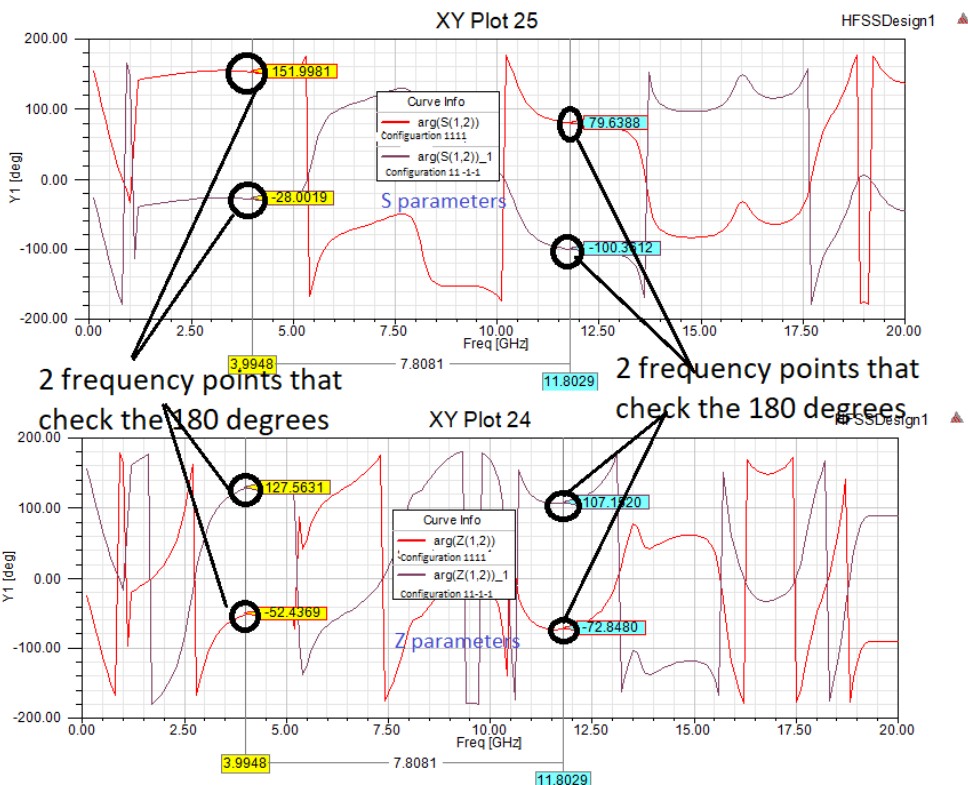

**Figure 9.** Arg (S12) and Arg (Z12) between two sources of four antennas in two different symmetry configurations 1111 and 11−1−1: The frequency range from 0 to 20 GHz . (Simulation under HFSS) (It checks $\Phi 2 - \Phi 1 = \pi$ (See after Equation (6)).

## 4. Advantage of Symmetry

Reducing the analysis domain is an advantage for symmetry problems [13,14,35]. In this section, we explain how symmetry reduces the computational domain of the used problem. Each symmetry plane reduces the computation by a factor of 2 [1–4]. Suppose that in our computation, we use two symmetry planes disposed axially along the two directions (ox) and (oy), for each symmetry state, which reduces the computation by a factor of four $(2 \times 2 = 2 + 2)$ [12–14]. According to the superposition theorem, each eigenstate of four antennas is the sum of four symmetry states [12–14], so in total, this reduces the calculation by a factor of 16 $(4 \times 4 = 4 + 4 + 4 + 4)$.

## 5. Conclusions

This paper presented the connection between the eigen-amplitudes of the four antennas and the symmetry of the amplitude of the associated sources (states) through mirror effects. A validation with the theory of symmetry point group is completed. In addition, a phase shift technique has been used to highlight all combinations of symmetry between electric and magnetic walls disposed along (ox) and (oy) axis. To distinguish these different symmetries, a method of calculating the S-parameters is introduced. Note that the main advantage of symmetry is to reduce the domain of analysis. As a perspective, we can apply this symmetry approach to (largely extended) antenna sub-arrays with different source amplitudes, using periodical walls outside of four antennas. So, it reduces more and more the computing time and memory space.

**Author Contributions:** Conceptualization, H.B.; methodology, H.B. and A.T.; software, H.B. and A.T.; validation, H.B. and A.T.; formal analysis, H.B.; investigation, H.B. and A.T.; resources, A.T.; data curation, H.B. and A.T.; writing—original draft preparation, H.B. and A.T.; writing—review and editing, H.B. and A.T.; visualization, H.B. and A.T.; supervision, A.T.; project administration, A.T.; funding acquisition, A.T. All authors have read and agreed to the published version of the manuscript.

**Funding:** This project received a part of the funding from the Laboratory Sys'Com-ENIT (LR-99-ES21)-National Engineering School of Tunis ENIT, Tunis, Tunisia, 1002.

**Institutional Review Board Statement:** Not applicable.

**Informed Consent Statement:** Not applicable.

**Data Availability Statement:** Not applicable.

**Acknowledgments:** The first part of this work is elaborated in collaboration with Henri Baudrand INP -N7 Toulouse. The authors thank Junwu TAO INP-N7 Toulouse for his help.

**Conflicts of Interest:** The authors declare no conflict of interest.

## Abbreviations

| | |
|---|---|
| MoM-GEC | Method of Moment combined with the Generalized Equivalent Circuits |
| HFSS | High-frequency structure simulator |
| CST | Computer Simulation Technology simulator |
| PEC | Perfect electric conductor |
| PMC | Perfect magnetic conductor |
| $C_{2v}$ | Symmetry point group of class $C_{2v}$ |

## Appendix A. Technique to Count the Phase Shift between Two Sources (Particular Cases of Odd and Even Symmetries)

This method explains how to calculate the phase between two sources. By the same reasoning established in [17], we imagine a line tracing placed at the input of a quadrupole of known parameter [S] (for example a source and its image), as proposed in Figure A1. This case is considered general to produce the phase shift between two sources by the addition of a line portion. This line segment provides a phase shift $\Phi1$ related to the propagation (in our case, related to the source ($S_1$) and its image ($S_2$)).

If we first assume that the output is matched, then $a_2 = 0$, and the input reflection coefficient undergoes two times the phase shift, so

$$S'_{11} = S_{11}e^{-(2j\Phi1)} \tag{A1}$$

The transmission coefficient from the input to the output undergoes the phase shift once, so

$$S'_{21} = S_{21}e^{-(j\Phi1)} \tag{A2}$$

If we now assume that the input is matched, then $a_1 = 0$, and the reflection coefficient seen from the output does not change

$$S'_{22} = S_{22} \tag{A3}$$

The reflection coefficient from the output to the input undergoes the phase shift only once, so

$$S'_{12} = S_{12}e^{-(j\Phi 1)} \tag{A4}$$

In short, it leads to

$$[S'] = \begin{pmatrix} S_{11}e^{-2j\Phi 1} & S_{12}e^{-j(\Phi 1)} \\ S_{21}e^{-j(\Phi 1)} & S_{22} \end{pmatrix} \tag{A5}$$

When the change concerns the two reference planes at the two accesses of a quadrupole, similar reasoning leads to (See Figure A2 [17]):

$$[S] = \begin{pmatrix} S_{11}e^{-j2\Phi 1} & S_{12}e^{-j(\Phi 2+\Phi 1)} \\ S_{21}e^{-j(\Phi 2+\Phi 1)} & S_{22}e^{-j2\Phi 2} \end{pmatrix} \tag{A6}$$

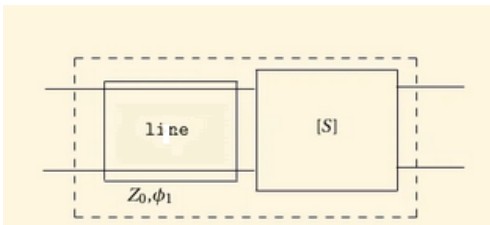

**Figure A1.** A portion of line added at the input of a quadrupole of known matrix [S] to reconstruct the phase established between 2 sources: we can work on the cases of symmetries as particular cases where the sources are in phase or opposition of phase) [17].

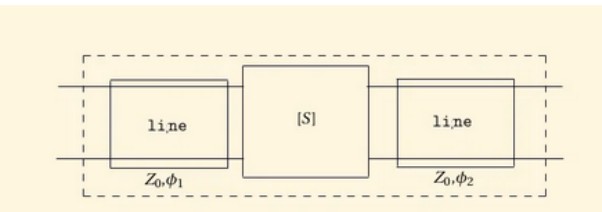

**Figure A2.** Line traces added to the input and output of a quadrupole of known matrix [S] [17].

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
