# Peer review of "Intuition and Symmetries in Electromagnetism: Eigenstates of Four Antennas"

_applsci, doi:10.3390/app122312049_

Round 1

Reviewer 1 Report

Intuition and symmetries in electromagnetism: Eigen states of antennas

By. Hamdi B. & Aguili T.

The authors presented several simulation examples and the connection between the Eigen-amplitudes of the 4 antennas along with the symmetry of the amplitude of the associated sources through mirror effects with the validation to the theory of symmetry point group.

Comments

(1)   How can you prove the symmetry in matrix (4)?

(2)   Punctuation is needed, for example after (7) “.”, also, P3/L72: in the first table, By normalizing theà in the first table. By normalizing the …

(3)   Avoid using block references, for example [24]-[33]. The authors must explain each gap in these references and then formulate the main contribution in their work.

(4)   Figs. Are not clear. All need more resolution and bigger font size and considering the template of the journal.

(5)   P4: delete   (See the last section)

(6)   It would be perfect if the authors added a comparison with an unsymmetrical case! 

I recommend it after the above considerations. 

Reviewer 2 Report

The examined paper is interesting and high-quality. The article’s content is relevant to the scientific area of the Applied Sciences Journal.

The article’s title represents the content and purpose of the article. The abstract is concise and relevant. The keywords are adequate. The Introduction and Abstract identify the need and relevance of this research. The article structure is clear. It contains explicitly required sections such as Introduction with references review, Problem Formulation, Results, Advantage of symmetry, and Conclusion. The paper contributes to the body of knowledge. The paper is technically sound. The provided references are applicable and sufficient.

There is one comment:

1) Is there any other limitation of the proposed idea? Such as more detailed limitations or specific ranges of the conditions should be explained. Please add more discussion and limitations of the proposed method.

There are some minor remarks:

1) No numerical results of the study are displayed in the Abstract and Conclusions.

2) In Figures 1-8, the inscriptions are very small and hard to be discerned.

3) Please check whether all notations in equations (2)-(7) were explained.

4) The future of this study should be added to the Conclusions.

In general, the article is good. Minor corrections are needed.

By considering the above, I recommend this paper for publication in the Applied Sciences Journal after minor revision.
